# Association between early methadone dose titration and treatment discontinuation and opioid toxicity: A retrospective cohort study

Ria Garg[1,2*], Jin Luo[1], Nikki Bozinoff[3,4], Beth Sproule[2,5,6], Tony Antoniou[1], Jennifer Wyman[4,7], Pamela Leece[4,8,9], Charlotte Munro[10], Jes Besharh[10], Tara Gomes[1,2,11,12]

1 ICES, Toronto, Ontario, Canada, 2 Leslie Dan Faculty of Pharmacy, University of Toronto, Toronto, Ontario, Canada, 3 Campbell Family Mental Health Research Institute, Centre for Addiction and Mental Health, Toronto, Ontario, Canada, 4 Department of Family and Community Medicine, University of Toronto, Toronto, Ontario, Canada, 5 Centre for Addiction and Mental Health, Toronto, Ontario, Canada, 6 Department of Psychiatry, University of Toronto, Toronto, Ontario, Canada, 7 Division of Addiction Medicine, Women's College Hospital, Toronto, Ontario, Canada, 8 Public Health Ontario, Toronto, Ontario, Canada, 9 University of Toronto Dalla Lana School of Public Health, Toronto, Ontario, Canada, 10 Ontario Drug Policy Research Network Lived Experience Advisory Group, Toronto, Ontario, Canada, 11 Li Ka Shing Knowledge Institute, St. Michael's Hospital, Toronto, Ontario, Canada, 12 Institute of Health Policy, Management and Evaluation, University of Toronto, Toronto, Ontario, Canada

* ria.garg@mail.utoronto.ca

**Data availability statement:** The dataset from this study is held securely in coded form at ICES. While legal data sharing agreements between ICES and data providers (e.g., healthcare organizations and government) prohibit ICES from making the dataset publicly available,

## Abstract

### Background

Clinical guidance in Canada and the United States recommends faster methadone dose titration for individuals using fentanyl. An initial dose increase is recommended between days 4 and 6, but the impact on patient outcomes remains unclear, particularly when provided in an outpatient setting. Therefore, we evaluated the association between early methadone dose titration (i.e., within treatment days 4–6) and treatment discontinuation and opioid toxicity.

### Methods and findings

We conducted a retrospective propensity-score weighted cohort study of Ontario residents who initiated methadone in an outpatient setting between January 2017 and December 2022. Exposure was defined as provision of a dose titration between treatment days 4–6, with the index date defined as the first date of dose increase. For unexposed individuals, the index date was randomly assigned and followed the same distribution as those exposed. Individuals were followed for up to 181 days following index date for study outcomes. Primary outcomes included methadone discontinuation and opioid toxicity, using an intention-to-treat approach. Secondary outcomes included methadone versus non-methadone-related toxicity while on treatment. We fit a propensity-score model to estimate the probability of

access may be granted to those who meet pre-specified criteria for confidential access, available at www.ices.on.ca/DAS (email: das@ices.on.ca). The full dataset creation plan and underlying analytic code are also available from the ICES via email to das@ices.on.ca upon request to qualified applicants, understanding that the computer programs may rely upon coding templates or macros that are unique to ICES and are therefore either inaccessible or may require modification.

**Funding:** Author T.G. received grants from the Ontario Ministry of Health (No. 0691) and the Canadian Institutes of Health Research (No. 496563), which supported this study. The funders had no role in study design, data collection and analysis, decision to publish, or preparation of the manuscript.

**Competing interests:** I have read the journal's policy and the authors of this manuscript have the following competing interests: N.B. reports receiving personal fees from Ontario College of Family Physicians for moderation of continuing professional development talks, some of which relate to opioid agonist therapy, outside the submitted work. T.G. reports consulting fees from Canada's Drugs Agency, honoraria from Indigenous Services Canada, and contracts with the Ontario College of Pharmacists and the Auditor General of British Columbia unrelated to this work. She has also been paid for expert testimony in an inquest led by the Ontario Office of the Chief Coroner. J.W. co-authored the methadone prescribing recommendation for people who use fentanyl, officially released by the Mentoring, Education, and Clinical Tools for Addiction – Partners in Health Integration in June 2021.

**Abbreviations:** CI, confidence interval; DDARD, Drug and Drug-Alcohol Related Death; ED, emergency department; HRs, hazard ratios; ICD, International Classification of Diseases; META-PHI, Mentoring, Education, and Clinical Tools for Addition – Partners in Health Integration; OAT, opioid agonist treatments; OUD, opioid use disorder; PHIPA, Personal Health Information Privacy Act; sIPTW, stabilized inverse probability of treatment weights; wHR, weighted hazard ratio.

dose titration, conditional on baseline demographic and clinical (e.g., comorbidities, past medication use) variables. Propensity score weighted Cox proportional hazard models were then estimated to determine the association between early dose titration and study outcomes. Among 13,560 incident methadone recipients (61.4% [$N = 8,322$] provided early dose titration), the mean age was 36.5 years old, and approximately one-third were female. Individuals who received an early dose titration had a lower weighted hazard of discontinuation, which was most pronounced during the first seven days of follow-up (weighted hazard ratio [wHR]: 0.55; 95% confidence interval [CI]: 0.51, 0.60), with attenuation towards the null over time. The observed weighted rate of opioid toxicity was lower among those exposed versus unexposed (10.1 versus 12.3 per 100 person-years; wHR: 0.82; [95% CI: 0.69, 0.97]). Early dose titration was not associated with opioid toxicity while on treatment. Most toxicities were attributed to non-methadone opioids (exposed: 3.89 per 100 person-years versus unexposed: 4.06 per 100 person-years; wHR: 1.00; [95% CI: 0.70, 1.44]). Methadone-related toxicity while on treatment remained low and comparable between exposed and unexposed groups (2.02 versus 2.65 per 100 person-years; wHR: 0.80; [95% CI: 0.46, 1.41]). The main study limitation is the lack of information on drug use history and time-varying clinical factors that may influence both dose titration and subsequent outcomes, introducing the potential for residual confounding.

## Conclusions

This population-based analysis demonstrated that early methadone dose titration was associated with improved treatment retention and lower hazard of opioid toxicity. Logistical barriers that hinder timely provision of dose increases must be addressed to improve methadone retention among people who use fentanyl.

## Author summary

### Why was this study done?

- Due to persistently low rates of methadone retention—partly attributed to increased opioid tolerance and dissatisfaction with traditional prescribing regimens among people who use fentanyl—clinical guidance in Canada and the United States has evolved to recommend more rapid methadone titration schedules.

- However, due to the urgency of the opioid crisis, these recommendations were made in the absence of robust clinical evidence. Ongoing concerns regarding the risk of methadone toxicity due to its highly variable pharmacokinetic profile remain.

## What did the researchers do and find?

- We used administrative data linked at the individual level to identify a cohort of Ontario residents who initiated methadone in an outpatient setting between January 2017 and December 2022.

- We examined associations between provision of an early methadone dose titration, defined as a dose increase between treatment days 4 to 6 and methadone discontinuation and opioid toxicity, compared with no dose change.

- In this retrospective propensity score-weighted cohort study of 13,560 incident methadone recipients, provision of an initial dose titration between treatment days four to six was associated with lower rates of methadone discontinuation and opioid toxicity over six months of follow-up, compared with no dose change.

## What do these findings mean?

- Logistical barriers that prevent people with high opioid tolerance from receiving timely dose titration need to be addressed given the association of early dose titration with improved methadone retention and lower rates of opioid toxicity.

- The study's main limitation was lack of comprehensive data on clinical indicators of opioid use disorder severity such as their drug use history.

## Introduction

The Canadian and United States' (US) fentanyl dominated unregulated drug supply has led to a 2-fold increase in the annual number of opioid related deaths between 2016 and 2023, rising from 45,081 to nearly 90,000 [1,2]. An estimated two-thirds of opioid-related deaths occur among people with opioid use disorder (OUD) [3,4]. While opioid agonist treatments (OAT) are safe and effective for people with OUD [5–8], treatment retention in the United States and Canada remains challenging as majority discontinue in the first few months [9–11]. This trend may reflect a mismatch between the high opioid tolerance of individuals exposed to a fentanyl-dominated unregulated drug supply and traditional OAT prescribing practices [9–12]. Recently released clinical guidance recognizes that people who use fentanyl require significantly higher methadone doses than were needed when the unregulated drug supply primarily consisted of heroin. For example, a focus group with people who use fentanyl reported that a minimum methadone dose of 120 mg was needed for adequate suppression of opioid cravings and withdrawal symptoms [13]. Due to requirements for higher methadone doses, evolving clinical guidance in the US and Canada recommends rapid dose titration to reduce time spent receiving a subtherapeutic dose and improve treatment retention [13–16].

In recent years, several US opioid treatment programs have adopted more aggressive titration schedules, with methadone doses increased daily, reaching doses of 70 mg or higher within the first week of treatment [17–19]. In Ontario, Canada, the 2021 guidance by Mentoring, Education, and Clinical Tools for Addition – Partners in Health Integration (META-PHI) recommended initiating methadone at 30 mg rather than 20–30 mg as recommended under the former provincial guidelines [20], with dose increases of 15 mg every 3–5 days until a dose of 75–80 mg per day is reached for people who use fentanyl [13]. Although both US and Canada face a highly potent and volatile unregulated drug supply, their methadone delivery model differs [21]. Specifically, in the US, methadone is exclusively prescribed and dispensed through federally certified opioid treatment programs, whereas in Canada, methadone is provincially regulated, commonly prescribed by outpatient clinicians (e.g., family physicians or nurse practitioners with an addiction medicine-focused practice) and dispensed through community pharmacies [21]. Because methadone prescribing and dispensing typically occur at separate locations in Canada, longstanding guidance recommends that prescribers conduct a follow-up assessment before increasing the dose to ensure lack of adverse effects (e.g., sedation).

Due to the urgency of the opioid crisis in Canada and US, recommendations for faster methadone dose titration were made in the absence of robust clinical evidence. Ongoing concerns remain about methadone accumulation and toxicity due to its highly variable pharmacokinetic profile [22]. Current guidance generally relies on clinical experience and emerging case reports from inpatient settings, where patients receive constant monitoring for toxicity and findings may not be generalizable to community-based care or opioid treatment programs [18,23–25]. Therefore, we investigated the association between provision of early methadone dose titration within an outpatient setting and subsequent methadone discontinuation and opioid related toxicity.

## Methods

### Study design and setting

We conducted a retrospective, population-based propensity score-weighted cohort study to examine the association between early methadone dose titration (i.e., within the first four to six days of treatment) and methadone discontinuation and opioid toxicity. We accrued Ontario residents who initiated methadone in an outpatient setting between January 1, 2017, and December 31, 2022, in Ontario, Canada. All individuals were observed for 181 days. This study was pre-registered (doi: osf.io/4v5f8) and adheres to STROBE reporting guidelines (S1 Table) [26].

### Data sources

We used administrative datasets housed at ICES, an independent, nonprofit research institute in Ontario with legal status that allows for the collection and analysis of administrative healthcare and demographic data without consent, for health system evaluation and improvement. The main data sources included the Narcotics Monitoring System, which captures all controlled substances (including methadone) dispensed from community pharmacies regardless of payer, and the Drug and Drug-Alcohol Related Death (DDARD) database, which provides post-mortem toxicology results for opioid-related deaths. Unique encoded identifiers were used to link the NMS to other datasets (see S2 Table for details on databases) and analyzed at ICES.

### Ethics statement

ICES is designated a prescribed entity under Ontario's Personal Health Information Privacy Act (PHIPA) and the Coroners Act. Section 45 of PHIPA authorizes ICES to collect personal health information without consent for the purpose of analysis or compiling statistical information with respect to the management of, evaluation or monitoring of, the allocation of resources to or planning for all or part of the health system. Data used in this study is authorized under section 45 of Ontario's Personal Health Information Protection Act and does not require review by a Research Ethics Board.

### Cohort

Our study cohort included all incident recipients of methadone meeting our inclusion and exclusion criteria. We defined incident methadone use as no recent dispensing record for OAT or immediate-release hydromorphone (e.g., safer opioid supply [27]) within a predefined lookback period, which varied according to the drug formulation. Specifically, we used a 30-day lookback period to define recent use of shorter-acting OAT (i.e., methadone, sublingual buprenorphine/naloxone, slow-release oral morphine) and immediate-release hydromorphone, and a lookback of 210 and 42 days for longer-acting buprenorphine formulations (i.e., buprenorphine implant and extended-release buprenorphine, respectively).

### Exposure definition

We identified all dispense records for methadone during the first six days of treatment. Individuals who received a dose increase during treatment days four to six (i.e., the exposure window) were defined as exposed, with the index date being

the first date the dispensed dose exceeded the initiation dose. For unexposed individuals (i.e., received the same dose for the first six days), the index date was randomly assigned and followed the same distribution as the number of days between initiation and index date among the exposed group.

### Exclusion criteria

We applied the following exclusion criteria on the date of methadone initiation to ensure incident use: methadone prescribed by an out-of-province practitioner, pharmacist, or dentist; provision of take-home doses; and methadone dose ≥60 mg. Next, we applied the following exclusions on the index date: individuals with missing patient identifiers, non-Ontario residents, recorded death date on or prior to the index date, age <18 or >64 years old, no record for methadone dispensing (for those unexposed), same day dispense record for another OAT product or immediate-release hydromorphone, hospital discharge within 14 days prior to index date, an emergency department (ED) visit on or between the initiation and index dates, provision of take-home doses, methadone prescribed by a non-physician or out-of-province prescriber, pregnancy, and an index date equal to treatment days 2–3 (i.e., before exposure window). Those who may have been prescribed methadone for pain, defined as individuals with a healthcare encounter for palliative care, cancer treatment, or cancer diagnosis in the year prior to the index date were also excluded. People not eligible for early dose titration were also excluded, which included those with a history of decompensated cirrhosis [28] in the prior year to index, those dispensed a lower dose on index versus initiation date, and people with three or more missed doses between the initiation and index dates. Individuals with missing values for any covariates included in our propensity score model were excluded from our analysis. Lastly, individuals could meet eligibility criteria at multiple points over the study period. Therefore, we randomly selected one incident treatment episode per individual to prevent dependence between observations.

### Outcomes and follow-up

Our primary outcomes were methadone discontinuation and fatal or non-fatal opioid toxicity. We defined methadone discontinuation as a gap in treatment of six or more days as this would require re-titration of methadone from its starting dose [13]. Coroner-confirmed fatal opioid toxicities were identified using the DDARD database. Nonfatal opioid toxicities were defined as any ED visit or inpatient hospitalization with a diagnosis of opioid toxicity (International Classification of Diseases [ICD] and Related Health Problems, 10th Revision diagnosis codes: T40.0-T40.4 and T40.6). As a secondary outcome, we examined whether the incident opioid toxicity event was attributed to methadone, defined using post-mortem toxicology reports or a healthcare encounter with an ICD-10 T40.3 code. We performed an intention-to-treat analysis for all primary outcomes, following individuals from their index date until the outcome, a non-opioid-related death (for opioid toxicity outcome), death date (for methadone discontinuation outcome), or end of follow-up (181 days or June 30, 2023). A per-protocol analysis for opioid toxicity was conducted to determine the hazard of these events while on treatment. In this analysis, we censored follow-up upon methadone discontinuation (with discontinuation date defined as five days following the last dispense date plus days' supply; see S3 Table for censorship indication). Finally, we conducted a stratified analysis to compare primary and secondary outcomes among participants who received a dose titration of <15 mg or ≥ 15 mg, relative to those unexposed.

### Statistical analysis

We fit a propensity-score model to estimate the probability of dose titration, conditional on baseline demographic (age, sex, income quintile, housing status, urban residence, Northern Ontario residence), clinical (comorbidities, healthcare encounters for alcohol-, benzodiazepine-, stimulant- and opioid-related toxicities, prior medication use), healthcare service use (non-OUD related outpatient visits, ED visits, hospital visits, attachment to primary care), methadone adherence (methadone dispense on the day prior to index, number of missed methadone doses between date of initiation and index, dose dispensed on initiation date, OUD-related outpatient visits between date of initiation and index) and prescriber

related variables (physician speciality, prescriber OAT client volume, years in clinical practice) defined on or prior to index date (see S4 Table for covariate definitions). We selected these covariates based on past literature and clinical judgement. We derived stabilized inverse probability of treatment weights (sIPTW) from the propensity score and trimmed the cohort at the 1st and 99th percentiles to reduce the influence of large values. We summarized baseline characteristics using descriptive statistics and compared exposure groups using weighted absolute standardized differences, with values >0.10 indicating imbalance. We used Cox proportional hazards models, weighted by the sIPTW, to compare outcomes between exposed and unexposed individuals. We verified the proportional hazards assumptions using time-varying exposure terms and visual inspection of Schoenfeld residuals. If the assumption was violated, we reported hazard ratios (HRs) over four intervals (0–7, 8–30, 31–90, and 91–181 days). Finally, we calculated sIPTW weighted rates per 100 person-years. We used 95% confidence intervals (CIs) to determine statistical significance, with estimates that crossed 1 considered statistically insignificant. All analyses were conducted at ICES using SAS Enterprise Guide 7.1.

### Sensitivity analysis

We conducted several sensitivity analyses to assess the robustness of our intention-to-treat models across different cohorts (i.e., those with no missed doses prior to index date and those whose index date was treatment day 4). Additionally, we redefined methadone discontinuation as a treatment gap of 14 or more days.

### Exploratory analysis

In an exploratory analysis (i.e., not included in pre-registered analysis), we examined the prevalence of dose titration after the index date among exposed and unexposed individuals, using the full cohort without application of propensity score trimming methods. We followed each individual for 14 days after their assigned index date to determine time to dose titration, defined as the first dispense date on which the dose exceeded the dose dispensed on index date. Lastly, we reconstructed the study cohort to include each individual's first continuous use period accrued during the exposure window and reanalyzed the primary study outcomes to test the robustness of our decision to randomly select one incident use period per person.

### Results

After application of our exclusion criteria, we included 13,836 individuals in our study cohort (Fig 1). In the primary analysis, following propensity score trimming, the cohort comprised 13,560 individuals, of whom 8,322 (61.4%) had their methadone dose titrated early (i.e., between treatment days four and six; Table 1). The mean age was 36.5 years (standard deviation: 10.3), approximately one-third (34.9%) were female and most (89.7%) resided in urban centers. Before weighting, individuals in the exposed group had a lower prevalence of past methadone use (26.7% versus 40.0%) and a lower mean initiation dose (23.8 mg versus 25.2 mg). Significantly higher measures of methadone adherence were noted among those exposed, as only 7.8% did not receive methadone the day prior to index and 3.2% missed two doses between initiation and index date, compared to 19.1% and 9.0%, respectively, among those unexposed. After propensity-score weighting, all baseline characteristics were balanced between exposure groups with standardized differences of <0.1 (S1 Fig) and good propensity score distributional overlap between those exposed and unexposed was observed (S2 Fig).

The provision of an early dose increase was associated with a lower weighted hazard of methadone discontinuation (weighted rate per 100 person-years: 208.9 versus 283.9; Table 2) and a longer median time to methadone discontinuation (109 [interquartile range: 23-inf] versus 63 days [interquartile range: 11-inf]) compared to those unexposed. The difference was most pronounced during the first seven days of follow-up (weighted hazard ratio [wHR] between days 0 and 7: 0.55; 95% CI: 0.51, 0.60) and attenuated towards the null over time (wHR from days 91 to 181: 0.89; 95% CI: 0.81, 0.99). See S5 Table for unadjusted event rates and HRs.

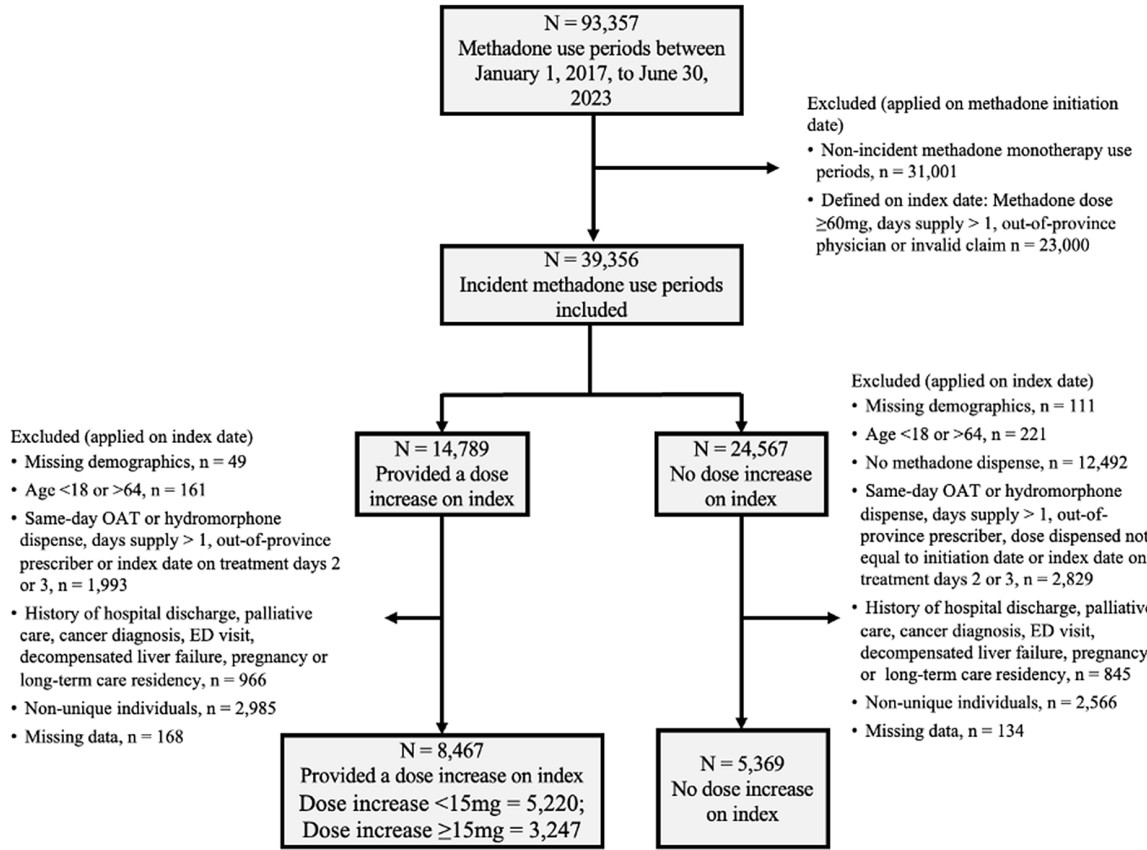

**Fig 1. Cohort exclusion flow diagram.**

In the intention-to-treat analysis, the observed rate of opioid toxicity was lower among exposed individuals (weighted rate per 100 person-years: 10.1 versus 12.3; wHR, 0.82 [95% CI, 0.69,0.97]). In the per-protocol analysis, which censored individuals at treatment discontinuation, respective opioid toxicity rates were lower in both exposure groups (5.91 versus 6.71 per 100-person-years) in comparison to the intention-to-treat analysis, and no association between early dose titration and opioid toxicity was observed (wHR: 0.93; 95% CI: 0.68, 1.26). Among events observed while on treatment, 66.2% (N = 149) were attributed to non-methadone opioids. There was no association between early dose titration and non-methadone toxicity (3.89 versus 4.06 per 100 person-years in exposed versus non-exposed groups, respectively; wHR 1.00; 95% CI 0.70, 1.44) or methadone toxicity (2.02 versus 2.65 per 100 person-years in exposed versus non-exposed groups, respectively; wHR 0.80; 95% CI: 0.46, 1.41).

## Stratified analysis

In the stratified analysis, after propensity score trimming, 5,123 individuals received a dose titration of <15 mg (median; interquartile range dose titration [IQR]: 10 mg; 10, 10; S6 Table), while 3,173 were titrated by ≥15 mg (median; IQR dose titration: 15 mg; 15, 15; S7 Table) on index date. While baseline characteristics were generally consistent with the overall analysis, those titrated by <15 mg had significantly lower prevalence of indicators suggestive of substance use disorder severity than those unexposed, including harmful or dependent stimulant use (11.7% versus 15.4%), injection-related infections (18.2% versus 22.3%), and prior opioid toxicity (8.1% versus 10.8%). After propensity-score weighting, baseline

**Table 1. Unweighted baseline characteristics of incident methadone recipients in Ontario, Canada, January 1, 2017 to December 31, 2022.**

| | Before weighting | | Standardized mean difference | |
|---|---|---|---|---|
| | **Unexposed (No dose titration; N = 5,238ᵃ)** | **Exposed (Dose titration; N = 8,322ᵃ)** | **Before weighting** | **After Weighting** |
| **Demographic Characteristics[b]** | | | | |
| **Age (Mean, SD)** | 36.1 (9.9) | 36.8 (10.5) | 0.07 | 0.01 |
| **Female** | 1,817 (34.7%) | 2,914 (35.0%) | 0.01 | 0.01 |
| **Income Quintile** | | | | |
| 1 | 2,042 (39.0%) | 3,130 (37.6%) | 0.03 | 0.01 |
| 2 | 1,263 (24.1%) | 2,027 (24.4%) | 0.01 | 0.01 |
| 3 | 931 (17.8%) | 1,401 (16.8%) | 0.02 | 0.01 |
| 4 | 597 (11.4%) | 1,037 (12.5%) | 0.03 | <0.01 |
| 5 | 405 (7.7%) | 727 (8.7%) | 0.04 | 0.01 |
| **Hospital flagged homelessness** (1 year prior) | 483 (9.2%) | 539 (6.5%) | 0.10 | 0.01 |
| **Low-income or disability support public drug plan** | 2,694 (51.4%) | 4,254 (51.1%) | 0.01 | 0.01 |
| **Residence in northern Ontario** | 592 (11.3%) | 828 (9.9%) | 0.04 | 0.02 |
| **Urban location of residence** | 4,705 (89.8%) | 7,464 (89.7%) | <0.01 | 0.01 |
| **Year of index date** | | | | |
| 2017 | 1,085 (20.7%) | 1,964 (23.6%) | 0.07 | 0.01 |
| 2018 | 871 (16.6%) | 1,492 (17.9%) | 0.03 | <0.01 |
| 2019 | 868 (16.6%) | 1,365 (16.4%) | 0.01 | <0.01 |
| 2020 | 877 (16.7%) | 1,430 (17.2%) | 0.01 | 0.01 |
| 2021 | 840 (16.0%) | 1,239 (14.9%) | 0.03 | <0.01 |
| 2022 | 690–695 | 825–830 | 0.10 | <0.01 |
| 2023 | ≤5 (0.1%) | ≤5 (0.0%) | 0.01 | 0.01 |
| **Comorbidities[b]** | | | | |
| **Charlson score** | | | | |
| No hospital visits | 3,566 (68.1%) | 5,594 (67.2%) | 0.02 | 0.01 |
| 0 | 1,378 (26.3%) | 2,253 (27.1%) | 0.02 | 0.01 |
| 1 | 204 (3.9%) | 316 (3.8%) | 0.01 | 0.01 |
| 2+ | 90 (1.7%) | 159 (1.9%) | 0.01 | <0.01 |
| **Human Immunodeficiency Virus** | 44 (0.8%) | 65 (0.8%) | 0.01 | 0.01 |
| **COPD** | 346 (6.6%) | 699 (8.4%) | 0.07 | <0.01 |
| **Asthma** | 1,285 (24.5%) | 2,121 (25.5%) | 0.02 | 0.01 |
| **Chronic Kidney Disease** (5 years prior) | 44 (0.8%) | 95 (1.1%) | 0.03 | 0.01 |
| **Liver Disease** (1 year prior) | 73 (1.4%) | 107 (1.3%) | 0.01 | 0.01 |
| **COPD related hospital or ED visit** (1 year prior) | 141 (2.7%) | 206 (2.5%) | 0.01 | <0.01 |
| **Asthma related hospital or ED visit** (1 year prior) | 35 (0.7%) | 54 (0.6%) | 0.00 | 0.01 |
| **Mental health related hospital or ED visit** (3 years prior) | 4,288 (81.9%) | 6,762 (81.3%) | 0.02 | <0.01 |
| **Psychotic disorders related outpatient visit** (3 years prior) | 769 (14.7%) | 1,071 (12.9%) | 0.05 | 0.01 |
| **Behavioral and neuro-developmental disorders related outpatient visit** (3 years prior) | 287 (5.5%) | 429 (5.2%) | 0.01 | 0.01 |
| **Other mental health disorders related outpatient visit** (3 years prior) | 863 (16.5%) | 1,264 (15.2%) | 0.04 | 0.01 |
| **Alcohol use disorder** (3 years prior) | 464 (8.9%) | 711 (8.5%) | 0.01 | <0.01 |

*(Continued)*

**Table 1.** (Continued)

| | Before weighting | | Standardized mean difference | |
|---|---|---|---|---|
| | Unexposed (No dose titration; N=5,238[a]) | Exposed (Dose titration; N=8,322[a]) | Before weighting | After Weighting |
| **Stimulant harmful use or dependence** (3 years prior) | 801 (15.3%) | 1,025 (12.3%) | 0.09 | <0.01 |
| **Sedative-hypnotic harmful use or dependence** (3 years prior) | 164 (3.1%) | 242 (2.9%) | 0.01 | <0.01 |
| **Hospital or ED visit for injection-related infection** (3 years prior) | 1,172 (22.4%) | 1,601 (19.2%) | 0.08 | <0.01 |
| **Hospital or ED visit for toxicity[b] (1 year prior)** | | | | |
| **Alcohol-related** | 22 (0.4%) | 30 (0.3%) | 0.01 | <0.01 |
| **Benzodiazepine-related** | 62 (1.1%) | 70 (0.8%) | 0.03 | <0.01 |
| **Stimulant-related toxicity** | 59 (1.1%) | 81 (0.9%) | 0.01 | 0.01 |
| **Non-fatal opioid toxicity** | 613 (11.1%) | 763 (8.8%) | 0.08 | <0.01 |
| **Healthcare utilization[b] (1 year prior)** | | | | |
| **Non-OUD related outpatient visits** | 4,522 (86.3%) | 7,341 (88.2%) | | |
| Mean (SD) | 7.5 (10.4) | 7.8 (10.7) | 0.06 | 0.01 |
| 0 | 716 (13.7%) | 981 (11.8%) | | |
| 1–4 | 2,157 (41.2%) | 3,406 (40.9%) | | |
| 5–10 | 1,195 (22.8%) | 1,944 (23.4%) | | |
| 11+ | 1,170 (22.3%) | 1,991 (23.9%) | | |
| **ED visits** | 3,055 (58.3%) | 4,841 (58.2%) | | |
| Mean (SD) | 2.1 (4.0) | 2.0 (4.0) | <0.01 | 0.01 |
| 0 | 2,183 (41.7%) | 3,481 (41.8%) | | |
| 1 | 1,062 (20.3%) | 1,758 (21.1%) | | |
| 23 | 1,048 (20.0%) | 1,662 (20.0%) | | |
| 4+ | 945 (18.0%) | 1,421 (17.1%) | | |
| **Hospital visit** | 601 (11.5%) | 912 (11.0%) | | |
| Mean (SD) | 0.2 (0.6) | 0.2 (0.6) | 0.02 | 0.01 |
| 0 | 4,637 (88.5%) | 7,410 (89.0%) | | |
| 1+ | 601 (11.5%) | 912 (11.0%) | | |
| **Attachment to Primary Care** | 4,703 (89.8%) | 7,570 (91.0%) | 0.04 | <0.01 |
| **Medication Use History[b]** | | | | |
| **Controlled Prescription Medication Use** (30 days prior) | 541 (10.3%) | 1,071 (12.9%) | 0.08 | <0.01 |
| Stimulants | 181 (3.5%) | 280 (3.4%) | 0.01 | 0.01 |
| Benzodiazepines | 430 (8.2%) | 889 (10.7%) | 0.08 | 0.01 |
| Non-OAT opioids | 0 (0.0%) | 0 (0.0%) | | |
| **Direct acting antivirals** (1 year prior) | 88 (1.7%) | 108 (1.3%) | 0.03 | <0.01 |
| **Opioid Agonist Treatment Use** (1 year prior) | 2,635 (50.3%) | 3,224 (38.7%) | 0.23 | 0.02 |
| Methadone | 2,093 (40.0%) | 2,222 (26.7%) | 0.28 | 0.01 |
| Buprenorphine/naloxone | 960 (18.3%) | 1,361 (16.4%) | 0.05 | 0.01 |
| Long-acting buprenorphine | 12 (0.2%) | 17 (0.2%) | 0.01 | <0.01 |
| Slow-Release Oral Morphine | 89 (1.7%) | 94 (1.1%) | 0.05 | <0.01 |
| **Immediate release hydromorphone** (1 year prior) | 117 (2.2%) | 255 (3.1%) | 0.05 | <0.01 |

*(Continued)*

**Table 1.** (Continued)

| | Before weighting | | Standardized mean difference | |
|---|---|---|---|---|
| | Unexposed (No dose titration; N = 5,238[a]) | Exposed (Dose titration; N = 8,322[a]) | Before weighting | After Weighting |
| **Methadone adherence characteristics following treatment initiation[b]** | | | | |
| **Dispense record for methadone the day before index date** | 4,237 (80.9%) | 7,673 (92.2%) | 0.34 | 0.01 |
| **Missed methadone doses between methadone initiation date and index date** | | | | |
| 0 | 3,448 (65.8%) | 7,233 (86.9%) | 0.51 | 0.02 |
| 1 | 1,317 (25.1%) | 822 (9.9%) | 0.41 | 0.01 |
| 2 | 473 (9.0%) | 267 (3.2%) | 0.24 | 0.01 |
| **Methadone dose on treatment initiation date** | | | | |
| Median (IQR) | 25 (20, 30) | 20 (20, 30) | | |
| Mean (SD) | 25.2 (7.5) | 23.8 (6.4) | 0.19 | 0.01 |
| **Methadone dose on index date** | | | | |
| Median (IQR) | 25 (20, 30) | 35 (30, 45) | | |
| Mean (SD) | 25.2 (7.5) | 37.0 (39.0) | | |
| **OUD-related outpatient visit between methadone initiation and index date** | 559 (10.7%) | 2,370 (28.5%) | 0.46 | 0.01 |
| **Physician Characteristics at Index[b]** | | | | |
| **Physician Specialty – Family Practitioner** | 4,081 (77.9%) | 6,605 (79.4%) | 0.04 | 0.02 |
| **Prescriber OAT Client Volume** | | | | |
| Low (lowest 50th percentile) | 614 (11.7%) | 732 (8.8%) | 0.10 | <0.01 |
| Moderate (51st to 80th percentile) | 1,621 (30.9%) | 2,758 (33.1%) | 0.05 | 0.01 |
| High (top 20th percentile) | 3,003 (57.3%) | 4,832 (58.1%) | 0.01 | <0.01 |
| **Years in Clinical Practice** | | | | |
| Mean (SD) | 23.3 (10.9) | 24.8 (10.7) | 0.14 | 0.01 |
| <10 years | 644 (12.3%) | 746 (9.0%) | 0.11 | |
| 10–19 years | 1,369 (26.1%) | 1,818 (21.8%) | 0.10 | |
| 20+ years | 3,225 (61.6%) | 5,758 (69.2%) | 0.16 | |

[a]Derived after applying propensity score trimming to the study cohort.

[b]Numbers represent n, (%), unless otherwise specified.

SD, standard deviation; ED, emergency department; OUD, opioid use disorder; OAT, opioid agonist treatment; COPD, chronic obstructive pulmonary disease; IQR, interquartile range.

characteristics were balanced between exposure groups in each stratified cohort (S1 Fig). Findings were consistent with the overall analyses among individuals titrated by <15 mg compared to those unexposed (S8 Table). Among individuals titrated by ≥15 mg versus those unexposed, the hazard of methadone discontinuation remained significantly reduced (wHR from days 1–7: 0.55, 95% CI: 0.50, 0.61); however, in contrast to our overall findings, we observed no association between early dose titration and opioid toxicity in the intention-to-treat analysis (wHR 0.95, 95% CI: 0.77, 1.16). For the per-protocol analysis, the proportional hazards assumption was violated and therefore we assessed the association at various times over follow-up. In this model, no association between early titration and opioid toxicity was observed in the first 90 days of treatment; however, we observed an elevated hazard of opioid toxicity between treatment days 91 and 181 (wHR 2.23, 95% CI: 1.14, 4.51) compared to unexposed individuals. This finding was driven by non-methadone-related toxicities (wHR from days 91 to 181: 3.29, 95% CI: 1.48, 8.07), whereas no association between methadone-attributed toxicity and dose titration was observed throughout follow-up (wHR 0.88, 95% CI: 0.46, 1.66).

**Table 2. Association between early dose titration and study outcomes.**

| Outcome | Rate[a] per 100 person-years (95% CI) | | Hazard ratio[a,b] (95% CI) |
|---|---|---|---|
| | Unexposed (No dose increase) | Exposed (Dose increase) | |
| **Methadone Discontinuation** | 283.9 (270.6, 297.9) | 208.9 (201.8, 216.2) | Interval 1[c]: 0.55 (0.51, 0.60) |
| | | | Interval 2[d]: 0.79 (0.73, 0.86) |
| | | | Interval 3[e]: 0.87 (0.80, 0.95) |
| | | | Interval 4[f]: 0.89 (0.81, 0.99) |
| **Opioid toxicity** | | | |
| Intention to treat | 12.3 (10.9, 14.0) | 10.1 (9.0, 11.2) | 0.82 (0.69, 0.97) |
| While on treatment | 6.71 (5.18, 8.69) | 5.91 (4.94, 7.07) | 0.92 (0.68, 1.26) |
| Methadone toxicity (while on treatment) | 2.65 (1.65, 4.28) | 2.02 (1.50, 2.73) | 0.80 (0.46, 1.41) |
| Non-methadone toxicity (while on treatment) | 4.06 (3.03, 5.42) | 3.89 (3.11, 4.86) | 1.00 (0.70, 1.44) |

[a]Stabilised inverse probability treatment weighting.

[b]Reference group: Unexposed.

[c]0 to 7 days of follow-up.

[d]8 to 30 days of follow-up.

[e]31 to 90 days of follow-up.

[f]91 to 181 days of follow-up.

CI, confidence interval.

## Sensitivity and exploratory analyses

Results from our sensitivity analyses were generally consistent with the main analysis. Findings for time to methadone discontinuation were robust to the application of sensitivity and exploratory analyses (S9 and S10 Tables). While the hazard of opioid toxicity did not significantly differ between exposure groups after application of cohort restrictions and selection of a unique continuous use period at random versus the first incident use period identified over the accrual period, toxicity rates remained lower among those exposed versus unexposed.

Overall, 68.1% of study participants received a dose increase within 14 days of the index date, with a median event time of 4 days (IQR: 3, 6 days) (S11 Table). The crude prevalence of post-index dose titration was significantly higher among those exposed versus unexposed (76.8% versus 54.4%; standardized difference: 0.49).

## Discussion

In this population-based study of incident methadone recipients in Ontario, Canada, individuals who received dose titration within an outpatient setting and in alignment with clinical guidance for people who use fentanyl had a lower hazard of methadone discontinuation and a lower or similar hazard of opioid toxicity over six months of follow-up compared to those who did not receive a dose titration during the first six days of treatment. Among individuals who remained on treatment, the rate of opioid toxicity was primarily driven by non-methadone opioids and was comparable between exposure groups.

A key finding of this study was that provision of an early dose titration (i.e., during the first four to six days of treatment) supported methadone retention without increased hazard of opioid toxicity among a cohort of individuals using a fentanyl-dominated unregulated drug supply. The safety and feasibility of rapid methadone titration is further supported by emerging evidence from US outpatient opioid treatment programs, with patients reaching doses up to 70 mg within one week of treatment without significant reports of toxicity and improved 30-day methadone retention [17,18]. Methadone is initiated at subtherapeutic doses due to its highly variable half-life and narrow therapeutic index, which places methadone

naïve individuals at risk of bioaccumulation and toxicity [22,29]. Currently, no clinical tools exist to determine the risk of an iatrogenic methadone toxicity at initiation, necessitating cautious titration [29]. However, individuals who use high-potency opioids—such as those in Canada and the US—have a high tolerance to opioids and are partially cross-tolerant to methadone, which may permit higher initiation doses and more rapid titration [13,16,30]. This is in agreement with our study results, conducted within a region where fentanyl dominates the unregulated drug supply, as early methadone dose titration remained safe even when the dose was titrated on treatment day 4 or by 15 mg with a median initiation dose of 30 mg.

A population-based study of methadone initiations in Ontario, Canada between 2015 and 2023 reported that only 40.5% received a dose titration within the first six days of treatment, with the prevalence of dose titration and methadone retention rates significantly declining over time [9,12]. In our study, even when restricted to those with no missed doses during the first six days of treatment, one-third of the cohort was not provided a dose increase. In Canada, methadone dose titration typically requires a follow-up physician visit, as standing orders are uncommon [13,16]. Whereas many opioid treatment programs in the US use standing induction titration orders covering treatment days 1–7, with nursing staff authorized to administer daily doses based on their assessment of sedation and intoxication. Requirements for a physician visit may contribute to many individuals remaining on initiation doses during the first week of treatment. For example, prior to weighting, we found the prevalence of an OUD-related physician visit between the initiation and index dates was significantly higher among those who received an early dose increase versus those unexposed. The first week of treatment with methadone can be challenging, as doses remain subtherapeutic and withdrawal symptoms and cravings persist, making it particularly difficult for patients to attend multiple appointments for dose adjustments [31]. To improve population-level methadone retention, physicians and patients should engage in shared decision-making at treatment initiation to develop a titration plan that allows for flexible reassessment, accounts for potential missed doses, and incorporates preplanned dose increases based on the patient's reported opioid tolerance—particularly for individuals who may be unable to see a physician for a dose increase during the first week of treatment, when discontinuation rates are highest. Future qualitative research should also explore barriers to dose titration to identify modifiable factors that may be addressed to improve titration rates and support safer, more effective methadone initiation.

Our study has several limitations. First, due to our use of administrative data, we did not have information on clinical factors (e.g., ongoing substance use) that may have evolved during the exposure window and influenced both the likelihood of dose titration and subsequent outcomes. Our findings are thus subject to residual and potentially time-varying confounding. Second, non-fatal opioid toxicities treated in the community without hospital transfer were not captured. Additionally, cases in which an opioid was not identified as the cause of toxicity in the patient's medical report upon ED or hospital admission may have resulted in missed non-fatal toxicity events. Therefore, our findings may underestimate the overall incidence of non-fatal opioid toxicity in this population. Third, given that defining the cause of a non-fatal toxicity relied on patients' self-reporting, toxicology testing, or both, there is potential for misclassification of non-fatal opioid toxicities that were attributed to methadone versus non-methadone opioids. Fourth, while we captured provision of the first dose titration, we did not evaluate outcomes related to subsequent dose titrations. Fifth, our study was conducted in a single Canadian province, with a distinct outpatient methadone delivery system, which may limit generalizability of our results to other jurisdictions. Sixth, we excluded treatment episodes initiated at doses ≥60 mg to ensure our cohort reflected true incident use and was aligned with current methadone prescribing guidance. While this may have excluded a small number of people starting methadone at very high doses, Canadian guidelines generally recommend a maximum starting dose of 40 mg in outpatient settings to mitigate the risk of methadone toxicity. Therefore, these cases most likely represent prevalent users whose prior dispensing history was not captured in our dataset (e.g., those transitioning from correctional facilities or moving to Ontario from other provinces). Lastly, as with all observational studies due to the potential for residual confounding, we cannot be certain of a causal association between exposure and study outcomes.

This population-based cohort study suggests that provision of an initial dose titration in alignment with the 2021 treatment guidance for people who use fentanyl in Ontario, Canada is associated with improved treatment retention and

reduced opioid toxicity rates. Logistical barriers that prevent people with high opioid tolerance from receiving timely dose titration, despite adequate adherence to methadone, should be addressed to improve treatment retention.

## Supporting information

**S1 Table. STROBE checklist of items that should be included in reports of cohort studies.**
(DOCX)

**S2 Table. Descriptions of all linked administrative databases used in the study.**
(DOCX)

**S3 Table. Censoring criteria applied to each outcome.**
(DOCX)

**S4 Table. Covariate Definition.**
(DOCX)

**S5 Table. Crude association between early dose titration and study outcomes.**
(DOCX)

**S6 Table. Baseline characteristics of incident methadone recipients in Ontario, Canada, January 1, 2017, to December 31, 2022, comparing no dose increase versus provision of a dose increase <15 mg.**
(DOCX)

**S7 Table. Baseline characteristics of incident methadone recipients in Ontario, Canada, January 1, 2017, to December 31, 2022, comparing no dose increase versus provision of a dose increase ≥15 mg.**
(DOCX)

**S8 Table. Association between early dose titration and study outcomes, stratified analysis.**
(DOCX)

**S9 Table. Association between early dose titration and study outcomes, sensitivity analysis.**
(DOCX)

**S10 Table. Association between early dose titration and study outcomes, exploratory analysis.**
(DOCX)

**S11 Table. Characteristics of dose titration following index date.**
(DOCX)

**S1 Fig. Absolute standardized differences before and after propensity score weighting.**
(TIFF)

**S2 Fig. Propensity score distribution between exposure groups.**
(TIFF)

## Acknowledgments

This study was supported by ICES, which is funded by an annual grant from the Ontario Ministry of Health (MOH) and the Ministry of Long-Term Care (MLTC). Parts of this material are based on data and information compiled and provided by the Ontario Ministry of Health, Ontario Health (OH), and CIHI. Data adapted from the Statistics Canada Postal CodeOM Conversion File was used, which is based on data licensed from Canada Post Corporation, and/or data adapted from the

Ontario Ministry of Health Postal Code Conversion File, which contains data copied under license from Canada Post Corporation and Statistics Canada. We thank IQVIA Solutions Canada for use of their Drug Information File.

The analyses, conclusions, opinions, and statements expressed herein are solely those of the authors and do not reflect those of the funding or data sources; no endorsement is intended or should be inferred.

## Author contributions

**Conceptualization:** Ria Garg, Nikki Bozinoff, Beth Sproule, Tony Antoniou, Jennifer Wyman, Pamela Leece, Charlotte Munro, Jes Besharh, Tara Gomes.

**Formal analysis:** Ria Garg, Jin Luo.

**Funding acquisition:** Ria Garg, Tara Gomes.

**Investigation:** Ria Garg, Tara Gomes.

**Methodology:** Ria Garg, Nikki Bozinoff, Beth Sproule, Tony Antoniou, Tara Gomes.

**Project administration:** Ria Garg, Tara Gomes.

**Resources:** Tara Gomes.

**Supervision:** Nikki Bozinoff, Beth Sproule, Tony Antoniou, Tara Gomes.

**Validation:** Ria Garg, Jin Luo, Charlotte Munro, Jes Besharh, Tara Gomes.

**Visualization:** Ria Garg, Nikki Bozinoff, Beth Sproule, Tony Antoniou, Jennifer Wyman, Pamela Leece, Charlotte Munro, Jes Besharh, Tara Gomes.

**Writing – original draft:** Ria Garg.

**Writing – review & editing:** Ria Garg, Jin Luo, Nikki Bozinoff, Beth Sproule, Tony Antoniou, Jennifer Wyman, Pamela Leece, Charlotte Munro, Jes Besharh, Tara Gomes.

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
