## [Editor Report · Decision Letter 0]

27 Aug 2025

Dear Dr Garg,

Thank you for submitting your manuscript entitled "Association Between Early Methadone Dose Titration and Treatment Discontinuation and Opioid Toxicity" for consideration by PLOS Medicine.

Your manuscript has now been evaluated by the PLOS Medicine editorial staff and I am writing to let you know that we would like to send your submission out for external peer review.

For clinical studies, please upload a copy of your trial study protocol as a supporting information file. The study protocol should be the version submitted for approval to the institutional review board or ethics committee, should include any amendments to the study protocol, as well as the date of their approval by the institutional review or ethics committee. Please also detail any deviations from the study protocol in the Methods section of your manuscript. The editors will consider the protocol and study conduct prior to a final decision for external review.

Please re-submit your manuscript within two working days, i.e. by Aug 29 2025.

Feel free to email me at atosun@plos.org or us at plosmedicine@plos.org if you have any queries relating to your submission.

Kind regards,

Alexandra Tosun, PhD

Senior Editor

PLOS Medicine

---

## [Decision Letter · Decision Letter 1]

29 Sep 2025

Dear Dr Garg,

Many thanks for submitting your manuscript "Association Between Early Methadone Dose Titration and Treatment Discontinuation and Opioid Toxicity" (PMEDICINE-D-25-02997R1) to PLOS Medicine. The paper has been reviewed by subject experts and a statistician; their comments are included below and can also be accessed here: [LINK]

As you will see, the reviewers find that your study addresses an important research question and is generally well-executed. They mostly ask for clarification and greater detail on treatment guidelines in Canada and the US. After discussing the paper with the editorial team and an academic editor with relevant expertise, I'm pleased to invite you to revise the paper in response to the reviewers' comments. We plan to send the revised paper to some or all of the original reviewers, and we cannot provide any guarantees at this stage regarding publication.

We ask that you submit your revision by Oct 20 2025. However, if this deadline is not feasible, please contact me by email, and we can discuss a suitable alternative.

Don't hesitate to contact me directly with any questions (atosun@plos.org).

Best regards,

Alexandra

Alexandra Tosun, PhD

Senior Editor

PLOS Medicine

atosun@plos.org

Comments from the academic editor:

This is a really interesting manuscript on an important topic for the North American context and must be appropriately framed in that regard. I think a much more global overview and contextualisation of the paper in the introduction and then discussion is necessary given that this is not an issue for all countries and contexts and the authors must be seen to be (and be) clear about that in the paper - this is an issue where highly potent opioids and likely very high opioid tolerance are the norm among illicit opioid users.

Comments from the reviewers:

Reviewer #1: Statistical review

This paper reports a retrospective analysis using propensity score weighting to investigate whether early methadone dose titration can improve outcomes compared to stable doses. Overall the statistical methods used are appropriate and the study is well reported. I had some comments on the paper, which are provided below:

1. Abstract: I would recommend that the start of follow-up is defined in the abstract.

2. Outcomes and follow-up: can the authors comment on the reliability of the outcome collection methods (especially the non-fatal ones).

3. Statistical analysis: I assume all the variables used in the PS model pre-date the exposure time? Some sounds ambiguous on that (e.g. number of missed methadone doses, hospital visits etc). Later on the results, e.g. table 1, do clarify this is done correctly. I would recommend the text here is updated.

4. Statistical analysis: I noted that it was mentioned missing data was grouped separately but I wasn't entirely sure what this meant. Can more be given on the missing data rates for variables and how these were dealt with in the PS model?

5. Statistical analysis: "A 2-sided type I error rate of 0.05 was used for all comparisons" - since p-values are not reported, I am not sure this statement is needed? It could be replaced by a statement about 95% confidence intervals being used.

6. Results "After propensity-score weighting, all baseline characteristics were balanced between exposure groups (eFigure 1)." - table 1 also shows the SMDs so I would mention here.

7. Results: it would be useful to visualise the propensity score distribution (or weights) by exposed/non-exposed group to demonstrate that there was overlaps, meaning the positivity assumption was met.

James Wason

Reviewer #2: Review of article: Association Between Early Methadone Dose Titration and Treatment Discontinuation and Opioid Toxicity

General comments -

This study examines an important area of methadone treatment that has had little exploration in the past - in spite of methadone being the most effective treatment for OUD for over 50 years, it remains unclear as to what is the most effective way to initiate and titrate the dose. While lower doses may decrease the potential risk of methadone related toxicity and overdose, they may lead to continued use of illicit opioids and early treatment discontinuation. This study is a good study that relates the dose titration early in methadone treatment episodes to outcomes - both retention in treatment and subsequent opioid toxicity, both from methadone and other opioids. This information will be of critical importance to patients and medical providers when they start on methadone. Overall, the study is well done, with a large database composed of many clinical and demographic measures. I would strongly support publication. Below are a few minor comments that might clarify some questions.

1) The paper briefly describes the protocols used in Canada and the US for initial methadone dose determinations in the discussion section. Most US readers will not be familiar with the Canadian system. I would suggest providing a more complete description in the introduction - this will help readers understand why you chose days 4-6 as the exposure window. US readers may not know that a physician visit is required for a dose increase and wonder why there aren't more dose increases in the first week. They may also not be familiar with how dosing is done and how take-home doses are determined.

2) The initial methadone dose on day 1 is given as a median - an average with calculation of standard difference would be more clarifying.

3) In figure 1, there were 93,357 "methadone use periods" - I assume this means this is the number of individuals who received any methadone in the time interval? Although some patients might be listed more than once due to multiple treatment episodes? Patients who were already on methadone (non-incident) were excluded. Why was a methadone dose >60 excluded - is this any dose throughout the period above 60 mg, or just during initiation? While this is mentioned in the text, it would help to put it on the figure as well.

4) Under Exclusion Criteria, you indicate that for individuals with more than one exposure, one of these was chosen randomly. Treatment retention declines with each subsequent treatment episode, so selecting from a group of episodes at random would have the effect of decreasing the average retention compared to the first recorded episode. I would consider selecting the earliest treatment episode rather than a random selection.

5) In Exclusion Criteria, you indicate that a methadone dose >60 mg was excluded (I assume for day 1, but it is not completely clear in the text), but don't indicate why this number was selected - the recommendation is for no more than 30 mg. Under what circumstances in Canada is the initial dose more than 30 mg, or more than 60 mg?

6) An indication of how take-home doses are determined in Canada is necessary to understand why this criterion was used. The exclusion was applied at initiation - does this mean only the initial dose? Over what period of time would a take-home dose be an exclusion? Does Canada allow take-home doses on Sundays and holidays (as in the US)?

7) Is there a good reason to exclude those <18 or >65 years old? This criterion is not indicated in Figure 1. How many were excluded on this basis?

8) You define a methadone discontinuation as missing 6 consecutive days of dosing. Why was this number chosen? Your definition of prior exposure used a 30-day window. Most OTPs in the US discharge patients after a 30-day dosing gap. US OTPs usually institute a dose reduction for a few days for a treatment gap of 6 days but would not consider this to be a readmission.

9) In the statistical analysis, the location of residence was used in the propensity adjustment, but it is not defined further in the text. Table 1 appears to indicate that this means either residence in northern Ontario or not in addition to rural vs. urban. What does it mean to be a resident of northern Ontario and why is it included?

10) In the first sentence of Statistical Analysis, you state that "prior medication use" is a demographic variable - which medication does this refer to?

11) The study uses the number of missed doses and a methadone dose dispensed prior to index in the propensity model. In the US, the likelihood of a dose increase on any day is strongly dependent on whether there have been any recent missed doses. A patient who misses a dose is unlikely to get dose increase on the following day. Patients who miss dosing during the first few days may differ from those who dose every day in important ways that are not completely addressed in the propensity adjustment. Would it be more accurate to simply exclude patients who missed doses prior to the index?

12) In the stratified analysis, for the per-protocol analysis, there was an elevated risk of opioid toxicity between days 91 - 180 for those titrated by 15 or more mg (only related to non-methadone opioids). This is not mentioned in the discussion - what would be a possible reason for this?

13) Under limitations, since this is an observational study, it is not possible to prove a causal connection between a dose titration and retention, only a statistical association after adjustment for all observed parameters. This is an important limitation of all non-randomized studies, and I think should be included in every case.

Reviewer #3: This paper describes a retrospective cohort study of methadone-treated patients, investigating the effects of early titration of the methadone dose. The research question is important, considering the on-going changes in patterns of opioid use, and the need for more data on different approaches to opioid agonist therapy. The manuscript is clearly written and easy to follow, and the basic methodology is sound (in the absence of experimental data). However, I have a couple of comments and questions:

1. The authors mention in the introduction that some treatment providers use a very fast induction schedule, sometimes with daily methadone dose titration. Nevertheless, the authors have chosen to define "early dose titration" as occurring from day four to six. What about the patients who may have received an even earlier dose titration, before day four?

2. I would be interested in knowing a bit more about the treatment of the unexposed patients, i.e. those that did not get any dose titration during the first six days. If I interpret eTable 7 correctly, 54.3% of them got a dose increase within 14 days of the index date (which corresponds to days 18-20 from treatment initiation). Isn't this a bit surprising? Where I work, we rarely wait more than a week before beginning the titration, and in this cohort almost half of the patients had to wait more than 2.5 weeks? Perhaps the requirement to meet a physician in order to change the dose, that is mentioned in the Discussion, might result in a sub-optimal dosing schedule even for some patients with "normal" titration rates? Maybe a comparison between early (<7 days), standard, and late (>14 days) titration starts would be of interest?

3. The authors mention a new guideline that was issued in 2021, recommending an early dose increase for people using fentanyl. However, the study period was from 2017 to 2022, which makes me wonder if the apparent effects of the exposure could be confounded by time, if most early titration cases occurred toward the end of the study period? I am not very familiar with the specific situation in Canada, but I imagine that changes in the drug market, as well as factors related to the covid-19-pandemic etc., could have resulted in different outcomes for this population over different parts of the study period.

In summary, this is an interesting study of an important clinical question. If the authors could clarify the questions above, I believe the manuscript could be suitable for publication.

---

* Please upload any figures associated with your paper as individual TIF or EPS files with 300dpi resolution at resubmission; please read our figure guidelines for more information on our requirements: http://journals.plos.org/plosmedicine/s/figures. While revising your submission, we strongly recommend that you use PLOS's NAAS tool (https://ngplosjournals.pagemajik.ai/artanalysis) to test your figure files. NAAS can convert your figure files to the TIFF file type and meet basic requirements (such as print size, resolution), or provide you with a report on issues that do not meet our requirements and that NAAS cannot fix.

After uploading your figures to PLOS's NAAS tool - https://ngplosjournals.pagemajik.ai/artanalysis, NAAS will process the files provided and display the results in the "Uploaded Files" section of the page as the processing is complete.

If the uploaded figures meet our requirements (or NAAS is able to fix the files to meet our requirements), the figure will be marked as "fixed" above. If NAAS is unable to fix the files, a red "failed" label will appear above.

When NAAS has confirmed that the figure files meet our requirements, please download the file via the download option, and include these NAAS processed figure files when submitting your revised manuscript.

* The funding statement should include: specific grant numbers, initials of authors who received each award, URLs to sponsors’ websites. Also, please state whether any sponsors or funders (other than the named authors) played any role in study design, data collection and analysis, the decision to publish, or preparation of the manuscript. If they had no role in the research, include this sentence: “The funders had no role in study design, data collection and analysis, decision to publish, or preparation of the manuscript.”

FIGURES AND TABLES

SUPPLEMENTARY MATERIAL

REFERENCES

STUDY TYPE-SPECIFIC REQUESTS

* Abstract: Please include the study design, population and setting, number of participants, years during which the study took place (enrollment and follow up), length of follow up, and main outcome measures.

* Please ensure that the study is reported according to the STROBE (or appropriate STOBE extension) guideline (available from: https://www.equator-network.org/reporting-guidelines/strobe) and include the completed STROBE (or STROBE extension) checklist as Supporting Information. Please add the following statement, or similar, to the Methods: "This study is reported as per the Strengthening the Reporting of Observational Studies in Epidemiology (STROBE) guideline (S1 Checklist)." When completing the checklist, please use section and paragraph numbers, rather than page numbers.

* Please check whether the RECORD guideline (available from https://www.record-statement.org) and include the completed checklist as Supporting Information might be more suitable for your study type.

* For all observational studies, in the manuscript text, please indicate: (1) the specific hypotheses you intended to test, (2) the analytical methods by which you planned to test them, (3) the analyses you actually performed, and (4) when reported analyses differ from those that were planned, transparent explanations for differences that affect the reliability of the study's results. If a reported analysis was performed based on an interesting but unanticipated pattern in the data, please be clear that the analysis was data driven.

* Please state in the Methods section whether the study had a prospective protocol or analysis plan. If a prospective analysis plan (from your funding proposal, IRB or other ethics committee submission, study protocol, or other planning document written before analyzing the data) was used in designing the study, please include the relevant document(s) with your revised manuscript as a Supporting Information file to be published alongside your study and cite it in the Methods section. A legend for this file should be included at the end of your manuscript. If no such document exists, please make sure that the Methods section transparently describes when analyses were planned, and when/why any data-driven changes to analyses took place. Changes in the analysis, including those made in response to peer review comments, should be identified as such in the Methods section of the paper, with rationale.

---

## [Decision Letter · Decision Letter 2]

18 Dec 2025

Dear Dr Garg,

Many thanks for re-submitting your manuscript "Association Between Early Methadone Dose Titration and Treatment Discontinuation and Opioid Toxicity" (PMEDICINE-D-25-02997R2) to PLOS Medicine. The paper has been seen again by the original subject reviewers and the statistician; their comments are included below and can also be accessed here: [LINK]

As you will see, the changes made to the paper were satisfactory to the reviewers. However, the academic editor provided several suggestions for additional analyses to strengthen the study, and we encourage you to include them in the manuscript. The academic editor also raised a concern about selecting a random treatment episode, and we have consulted the statistical reviewer on this issue. Please see their comments below. After discussing the feedback and the paper with the editorial team, I'm pleased to invite you to revise the paper in response to the editors' comments. We plan to send the revised paper to some or all of the original reviewers, and we cannot provide any guarantees at this stage regarding publication.

When you upload your revision, please include a point-by-point response that addresses all of the editorial points, indicating the changes made in the manuscript and either an excerpt of the revised text or the location (eg: page and line number) where each change can be found. Please also be sure to check the general editorial comments at the end of this letter and include these in your point-by-point response. When you resubmit your paper, please include a clean version of the paper as the main article file and a version with changes tracked as a marked-up manuscript. It may also be helpful to check the guidelines for revised papers at http://journals.plos.org/plosmedicine/s/revising-your-manuscript for any that apply to your paper.

We ask that you submit your revision by Jan 08 2026. However, if this deadline is not feasible, please contact me by email, and we can discuss a suitable alternative.

Due to the upcoming holiday season, the journal will operate at reduced capacity from December 22 to January 2. This will cause delays in the manuscript process. We appreciate your understanding.

Don't hesitate to contact me directly with any questions (atosun@plos.org).

Best regards,

Alexandra

Alexandra Tosun, PhD

Senior Editor

PLOS Medicine

atosun@plos.org

Comments from the academic editor:

The authors made a conclusion that high dose methadone (>60mg) should be excluded as a starting dose and yet it is high dose inductions that is the focus of the study - I wonder whether the authors might need to do some additional analysis - albeit post hoc - to consider a) whether inclusion of the high doses they have excluded changes the observed findings, b) whether the high dose initiations look different to the lower dose ones in terms of characteristics of the people or treatment episodes (i.e. any basis for their assumption that these are episodes continued from out of jurisdiction or incarceration) and c) whether there is a time pattern observed whereby proportionally more of these occur in later years of the study - since dose is the object of focus of the study and this is a significant limitation in capture of the dataset. I think it would be helpful to do this additional investigation.

Additionally, the justification for selecting a random treatment episode (rather than the first) does not seem entirely robust. Selecting a random event means that you will randomly usually select a subsequent episode not a first one - at least if you use the first observed episode, you're minimising the number of times where you are using a repeated treatment entry that the authors claim would be at play given the truncation of the observation period?

Editorial note: We consulted the statistical editor regarding the second comment, and they agree with the academic editor's point. They agree that your justification for using random selection instead of an alternative method (e.g., the first eligible visit) is insufficient. They suggest conducting a sensitivity analysis using an alternative method, such as the first visit; we strongly encourage you to perform this analysis.

Comments from the reviewers:

Reviewer #1: Thank you to the authors for fully addressing my previous review comments. I have no further issues to raise.

Reviewer #2: Thank you for your detailed responses. Thank you also for your contribution to this important area. I have no more comments or revisions at this time.

Reviewer #3: The authors have made a thorough revision of the manuscript, with a number of changes that made it stronger and improved clarity. I agree with the other reviewer comments, that it is important to contextualize the findings, for readers like me that are less familiar with the specifics of the Canadian treatment system. I believe that this version of the manuscript makes this more clear, and I have no further comments to add at this point.

Requests from Editors:

GENERAL

* Please confirm that your title complies with to PLOS Medicine's style. Your title must be nondeclarative and not a question. It should begin with main concept if possible. "Effect of" should be used only if causality can be inferred, i.e., for an RCT. Please place the study design ("A randomized controlled trial," "A retrospective study," "A modelling study," etc.) in the subtitle (ie, after a colon).

* Statistical reporting: Please revise throughout the manuscript, including tables and figures.

- Please report statistical information as follows to improve clarity for the reader ""22% (95% CI [13,28]; p</=)"".

- Please separate upper and lower bounds with commas instead of hyphens as the latter can be confused with reporting of negative values.

- Please repeat statistical definitions (HR, CI etc.) for each set of parentheses.

* Please ensure that all abbreviations are defined at first use throughout the text (including statistical abbreviations).

* Please ensure that tables and figures, including those in supplementary files, are appropriately referenced in the main text.

* Please review your text for claims of novelty or primacy (e.g. 'for the first time' or ‘novel’) and remove this language.

* Please confirm that any use of statistical terms (such as trend or significant) are supported by the data, and if not please remove them. The term trend should be used only when the test for trend has been conducted.

* Please define all acronyms used in each figure or table in its corresponding legend.

* Please confirm the use of patient-centered language. Please note that patient-centered language is constructed with the use of post-modified nouns putting the person first in the sentence structure.

* Your study is observational and therefore causality cannot be inferred. Please check your manuscript and confirm that you have not used language that implies causality.

* Please consider including an Acknowledgments section in your manuscript, acknowledging the study participants as well as the individuals who played a role in data collection or participant care or other involvement.

ABSTRACT

* Please confirm that your abstract complies with our requirements, including providing all the information relevant to this study type https://journals.plos.org/plosmedicine/s/submission-guidelines#loc-abstract

* Please confirm that all numbers presented in the abstract are present and identical to numbers presented in the main manuscript text.

* In the abstract, please include the important dependent variables that are adjusted for in the analyses.

AUTHOR SUMMARY

* “Among people who use a fentanyl-dominated unregulated drug supply, the provision of an early methadone dose titration supported methadone retention, which appears to have led to reduced opioid toxicity.” – please avoid claims of causality.

* In the author summary, in the final bullet point of 'What Do These Findings Mean?', please include the main limitations of the study in non-technical language.

METHODS AND RESULTS

* Thank you for providing your STROBE checklist. Please replace the page numbers with paragraph numbers per section (e.g. "Methods, paragraph 1"), since the page numbers of the final published paper may be different from the page numbers in the current manuscript.

* Please confirm that you provided the unadjusted comparisons as well as the adjusted comparisons in all relevant Tables.

* Please confirm that you specified the variables controlled for in all relevant Tables.

DISCUSSION

* Please remove the 'conclusions' subheading from the discussion. Please also remove any other subheadings from the discussion.

General Editorial Requests

---

## [Decision Letter · Decision Letter 3]

19 Feb 2026

Dear Dr Garg,

On behalf of my colleagues and the Academic Editor, Louisa Degenhardt, I am pleased to inform you that we have agreed to publish your manuscript "Association Between Early Methadone Dose Titration and Treatment Discontinuation and Opioid Toxicity: A retrospective cohort study" (PMEDICINE-D-25-02997R3) in PLOS Medicine.

I appreciate your thorough responses to the reviewers' and editors' comments throughout the editorial process. For transparency, I am sharing the statistical reviewer's feedback on the sensitivity analyses below my signature. We look forward to publishing your manuscript, and editorially there are only a few remaining points that should be addressed prior to publication. We will carefully check whether the changes have been made. If you have any questions or concerns regarding these final requests, please feel free to contact me at atosun@plos.org.

Please see below the minor points that we request you respond to:

* Table 1: Please indicate below the table that the numbers are "n, (%)" unless otherwise specified.

* l.316, “and a longer median time to methadone discontinuation (108 versus 63 days) compared to those unexposed.” – Are these numbers available in a table? Please clarify.

* Sensitivity analyses: We suggest adding a statement explaining that no differences in opioid toxicity were observed for both analyses.

Before your manuscript can be formally accepted you will need to complete some formatting changes, which you will receive in a follow up email (including the editorial requests above). Please be aware that it may take several days for you to receive this email; during this time no action is required by you. Once you have received these formatting requests, please note that your manuscript will not be scheduled for publication until you have made the required changes.

PRESS

Sincerely,

Alexandra Tosun, PhD

Senior Editor

PLOS Medicine

Comments from Reviewers:

Reviewer #1: Thank you to the authors for adding the additional sensitivity analysis looking at choice of eligible visit. It is reassuring that the results are robust to this alternative choice.